# 25(OH)D Concentration in Neonates, Infants, Toddlers, Older Children and Teenagers from Poland—Evaluation of Trends during Years 2014–2019

**DOI:** 10.3390/nu15153477

**Published:** 2023-08-06

**Authors:** Marek Wójcik, Maciej Jaworski, Paweł Płudowski

**Affiliations:** Department of Biochemistry, Radioimmunology and Exerimental Medicine, The Children’s Memorial Health Institute, 04-730 Warsaw, Poland; m.wojcik@ipczd.pl (M.W.); m.jaworski@ipczd.pl (M.J.)

**Keywords:** vitamin D, 25(OH)D, vitamin D deficiency, calcemia, infants, toddlers, children, adolescents

## Abstract

Introduction: Local and international guidelines have provided schedules for the vitamin D supplementation of general populations of different ages, including children. Our study aimed to assess 25(OH)D concentration and its potential change during a growth and maturation period, adding parameters that reflect the risk of hypercalcemia. Materials and methods: The available 25(OH)D concentration values (*n* = 17,636; 7.8 ± 6.0 years), calcium (*n* = 2673; 16.3 ± 6.1 years) and phosphate (*n* = 2830; 3.8 ± 5.2 years) metabolism markers were analyzed in a studied group of patients (0–18 years). Results: In the studied group the mean 25(OH)D concentration was 29.4 ± 11.7 ng/mL. Concentrations of 25(OH)D < 10 ng/mL were observed in 1.7% of patients (*n* = 292), 10–20 ng/mL in 17.2% (*n* = 3039), 20–30 ng/mL in 39.5% (*n* = 6960) and 30–50 ng/mL in 37.2% (*n* = 6567). In patients with a 25(OH)D concentration <10 ng/mL, normal calcemia (2.25–2.65 mmol/L) was observed in 29.5% of cases (*n* = 86). Three patients had 25(OH)D concentrations above 100 ng/mL with co-existing hypercalcemia; the mean was Ca = 3.40 mmol/L. Hypocalcemia (Ca < 2.25 mmol/L) was observed in 10,4% of patients (*n* = 2797). Furthermore, 5.0% of patients showed an increased calcium concentration >2.65 mmol/L (*n* = 1327). The highest mean 25(OH)D concentration of 32.1 ng/mL ± 12.9 was noted in the years 2018–2019 (*n* = 3931) and the lowest in the year 2015 (27.2 ng/mL ± 11.0; *n* = 2822). Conclusions: Vitamin D deficiency (<20 ng/mL) was noted in 18,9% of subjects in the years 2014–2019. An effective prevention of vitamin D deficiency was observed in children aged 3 years and younger. A relationship between the concentrations of calcium and 25(OH)D was not observed.

## 1. Introduction

Wide health benefits resulting from maintaining the optimal concentration of vitamin D have been confirmed by many scientific publications over the last several years.

Vitamin D deficiency was associated with or coincided with many civilization diseases, such as cancer [1,2,3,4,5], autoimmune diseases [6,7,8], diabetes type 1 and 2 [9,10,11], cardiovascular disease [12,13,14], hypertension [15,16,17] and depression, schizophrenia [18,19] or Alzheimer’s disease [20]. Maintaining proper serum 25-hydroxyvitain D, i.e., 25(OH)D, became a serious problem and appeared to be an important aspect for human health [21,22,23,24,25,26]. Earlier, the criterion of vitamin D deficiency associated with the risk of deficiency rickets or osteomalacia was considered when a 25(OH)D concentration value below 10 ng/mL was noted in a given patient [27]. However, in the last 10 years, many world scientific and medical societies have published guidelines on the recommended values of 25(OH)D-a metabolite recognized as an indicator of a vitamin D deficiency. The Institute of Medicine (IOM; Washington, DC, USA) recognized the value of 20 ng/mL as the minimum concentration of 25(OH)D necessary to obtain health benefits [28]. The Endocrine Society (ES; Washington, DC, USA) recommended a minimum concentration of 25(OH)D > 30 ng/mL, and indicated the optimal range as being 40–60 ng/mL [29]. In addition, the International Osteoporosis Foundation recommended a value of 30 ng/mL pointing to an important correlation with optimal parathormone (PTH) concentration values [30].

Many other societies around the world have also developed their own guidelines. Comparing these data, it is possible to notice differences in the proposed target values. This is due to differences of opinion as to the perception of the effects of vitamin D by different groups of experts. The IOM recommendations focused on benefits limited to bone tissue and calcium–phosphate metabolism. The ES proposal pointed to the systemic effect of vitamin D. The ES defined values <30 ng/mL and <20 ng/mL as a “vitamin D insufficiency” or a “deficiency”, respectively.

The recommendations for Europe as well as for the Scandinavian countries, Germany, Austria and Switzerland, adopted a sufficient concentration of 25(OH)D as being at least 20 ng/mL [31,32,33,34].

In 2013, the guidelines for Central Europe were published for a healthy population and for the groups at risk of deficits. The recommended daily doses of vitamin D, the diagnostic criteria for 25(OH)D and the recommendations regarding maximum safe doses were published [35]. In this document, a minimum target concentration of 25(OH)D of 30–50 ng/mL was proposed for both the calcemic and pleiotropic effects of vitamin D [35]. In 2018 [36] and 2023, [37] updates were introduced with national, i.e., Polish, arrangements and guidelines. The detailed rules for vitamin D supplementation and treatment were described, the global and local recommendations on supplementation were compared and safety issues were discussed.

The symptoms of the toxic effect of vitamin D were shown to be extremely rarely observed, and accompanying 25(OH)D concentrations of up to 100 ng/mL had been established as safe for children and adults [37,38].

The problem of potential toxicity concerns only people with congenital hypersensitivity to vitamin D, including a group of genetically predisposed people, in whom the supply of even the recommended doses of vitamin D, considered as safe, may introduce health problems due to hypercalcemia and hypercalciuria [39]. The risk of vitamin D intoxication, especially in unrecognized newborns, infants and toddlers with idiopathic infantile hypercalcemia (IIH) or with the Williams–Beuren syndrome, granulomatous diseases and lymphomas should be considered before the start of vitamin D supplementation [40,41,42,43,44,45,46,47,48].

This study follows and continues the article “25(OH)D Concentration in Neonates, Infants, and Toddlers from Poland–Evaluation of Trends During Years 1981–2011” [49], where we have retrospectively assessed the parameters of the Ca-P metabolism and 25(OH)D concentration among a population aged 0–18 years, but investigated in the previous period of time. Again, the 25(OH)D concentration data as well as the calcium–phosphate metabolism markers assayed in the years 2014–2019 were analyzed to investigate the scope of potential changes.

## 2. Patients and Methods

### 2.1. The Study Group

The study group consisted, in total, of 49,424 pediatric cases aged 0–18 years with a mean age of 7.3 ± 6.0 years. A total of 17,636 patients from this group had results for a 25(OH)D concentration value, 26,331 had results for a calcium (Ca) in serum concentration value and 2839 had results for a parathormone (PTH) value. The study group represented the entire pediatric population of Poland, healthy or with congenital and acquired disorders of Ca-P homeostasis. The medical database analyzed in this study consisted of results assayed in our laboratory in the years 2014–2019. The inclusion criterion was access to the 25(OH)D measurement value, and/or the Ca and PTH values of a patient’s sample. All lab assays were conducted at the same time.

### 2.2. Methods

Biochemical parameters were assessed using the following methods: (1)Total calcium (Ca) was assayed with the use of spectrophotometric assay (Alinity Abott);(2)25-hydroxyvitamin D (25(OH)D) and 1,25-dihydroxyvitamin D (1,25(OH)2D) concentration values were assessed using chemiluminescence immunoassay CLIA (IDS-ISYS);(3)Parathormone (PTH) activity measurements were conducted by immunoradiometric assay IRMA (Cisbio Bioassays).

The above-mentioned methods did not changed during the years that were included in this study, i.e., 2014–2019. The authors had access to The Children’s Memorial Health Institute’s medical database for the laboratory assay results that were evaluated.

### 2.3. Statistical Analyses

The medical database was investigated in terms of patient’s age, year of visit, quarter of the year of assay and 25(OH)D concentration value and/or Ca value or PTH value or 1,25(OH)_2_D value. The mean ± standard deviation values of 25(OH)D, Ca and PTH concentrations were calculated as general characteristics of the studied group. The distribution of the concentration values of 25(OH)D, Ca, PTH in relation to “subjective” reference values was also calculated. In cases with evident hypercalcemia, i.e., Ca ≥ 2.75 mmol/L, both 25(OH)D and 1,25(OH)_2_D as well as the PTH distribution of values were investigated. Finally, 25(OH)D concentration values were evaluated according to patient’s age, year of the visit and quarter of the year, and correlations between a measure of vitamin D status and patient’s age or Ca concentrations were estimated; mean ± standard deviation values and, if statistically significant, *r* values were provided. All the statistical analyses were carried out with the use of Statistica v. 10 (StatSoft Inc., Tulsa, OK, USA).

## 3. Results

### 3.1. General Characteristic of the Study Group

The means of the Ca, 25(OH)D and PTH concentration values were calculated based on the accessible data from the medical database. The number of patients aged 0–18 years was 49,424 (mean age 7.3 ± 6.0 years). Among this group, 25(OH)D concentration values were available in a group of 17,636 patients (mean age 7.8 ± 5.8 years). In this group, the mean 25(OH)D concentration value was 29.4 ± 11.7 ng/mL (Table 1). The means of biochemical parameters were within the reference ranges for a given age (Table 1).

### 3.2. Biochemical Characteristics

For 292 cases (<2%), 25(OH)D concentrations < 10 ng/mL were noted, values of 10–20 ng/mL were observed in 3039 cases (17%), values > 20–30 ng/mL were observed in 39.5% of cases (*n* = 6960) and values > 30–50 ng/mL were found in 37% (*n* = 6567) (Table 2). A 25(OH)D concentration value of higher than 125 ng/mL was noted only in four cases including one case with evident hypercalcemia (Ca = 4.82 mmol/L).

Among the group with severe vitamin D deficiency, defined as having a 25(OH)D concentration <10 ng/mL (*n* = 119), serum Ca concentrations were normal (2.25–2.65 mmol/L) in 72.3% of the patients (*n* = 86). Thiry one patients (26.1%) were identified with hypocalcemia (Ca <2.25 mmol/L) and two patients (1.6%) showed Ca levels above 2.65 mmol/L. In those subgroups, the mean values of 25(OH)D were 7.0 ng/mL and 8.8 ng/mL, respectively.

In the subgroup with 25(OH)D concentrations exceeding 100 ng/mL, the concentrations of serum Ca varied between 1.92 and 4.82 mmol/L (on average 2.48 mmol/L; *n* = 26). The percentage of patients showing an increased 25(OH)D concentration with coexisting hypocalcemia (Ca < 2.25 mmol/L) was 15.3% (*n* = 4) and the percentage with elevated Ca levels (>2.65 mmol/L) was7,7% (*n* = 2). The respective 25(OH)D values among these two patients were 103.7 ng/mL and 125.0 ng/mL.

The minimal Ca concentration value noted in the whole group was 1.03 mmol/L. This value coincided with a 25(OH)D concentration of 15.3 ng/mL. The highest Ca value was a 4.82 mmol/L with a 25(OH)D concentration of 125 ng/mL.

Table 3 presents the 25(OH)D concentration values in cases with evident hypercalcemia, defined as Ca ≥ 2.75 mmol/L. Evident hypercalcemia was observed in 0.35% (*n* = 93) of the entire population studied in our research. 

In the subgroup of patients with evident hypercalcemia, most of them (*n* = 48, 51.6%) presented an optimal concentration of 25(OH)D (>30–50 ng/mL)—or a suboptimal concentration (>20–30 ng/mL) (*n* = 21, 22.6%). Fourteen hypercalcemic patients revealed high 25(OH)D concentrations (50–100 ng/mL). Only 1 person showed a potentially toxic concentration of 25(OH)D–125.0 ng/mL. The prevalence of evident hypercalcemia was not significantly related to the 25(OH)D concentration range. The average Ca concentration value in the group with evident hypercalcemia was 2.84 ± 0.16 mmol/L. The correlations between Ca and 25(OH)D concentrations were statistically not significant both in the entire group or in hyper- or hypocalcemic subgroups.

Analysis of the 1,25(OH)_2_D concentration values in the subgroup with evident hypercalcemia (Table 4) showed that in more than half of patients (54.1%), the concentration was above 80 pg/mL, described in clinical practice as the upper limit of the reference values. 

Table 5 presents the PTH concentration values in the subgroup of patients with evident hypercalcemia (Ca ≥ 2.75 mmol/L). More than half of all patients (53.7%) have not reached the lower limit of reference values −11 pg/mL (reference values 11–62 pg/mL according to Cisbio Bioassays).

### 3.3. 25(OH)D Concentration According to Age

The two younger groups relative to the older groups revealed higher means of 25(OH)D concentration (Table 6). The highest mean 25(OH)D concentration of 36.3 ng/mL ± 14.2 was observed in infants aged 0–12 months (*n* = 3483). The lowest mean 25(OH)D concentration of 25.4 ng/mL ± 9.1 was noted in children of up to 10 years (*n* = 6908). The correlation between age and 25(OH)D concentrations was significant and appeared to be negative (R spearman = −0.364445, *p* < 10–17), as shown in Figure 1.

### 3.4. 25(OH)D Concentration in the Time Periods

Table 7 presents the means of 25(OH)D concentration in studied time periods. The highest mean 25(OH)D concentration was noted in the years 2018–2019 (*n* = 11,319) and reached 32.1 ± 12.9 ng/mL. The lowest mean 25(OH)D concentration was observed in 2015 (27.2 ± 11.0 ng/L).

### 3.5. 25(OH)D Concentrations According to Season of the Year

The 25(OH)D concentration in the autumn–winter season (I-IV quarter) was lower than in the spring–summer season (II-III quarter), which was expected (Table 8).

## 4. Discussion

There are many reports providing vitamin D recommended intakes, reference ranges for 25(OH)D and potential toxicity risks [35,36,37,50,51]. Other reports confirmed that vitamin D deficiency is a serious healthcare problem [25,26,27,28,29,30,52].

In our earlier study titled “25(OH)D Concentration in Neonates, Infants, and Toddlers from Poland–Evaluation of Trends During Years 1981–2011” [49] we showed a significantly higher average 25(OH)D concentration (37.5 ng/mL) in the group of newborns and infants (0–18 month) compared to children and adolescents (18 month–18 years of life) [49]. A clear negative trend showing decreasing 25(OH)D concentrations with age was revealed, especially after the age of 3 [49]. In this paper, we decided to present another time period, i.e., the evaluation of 25(OH)D concentration values in the years 2014–2019. Our study was based on a representative group of 49,424 pediatric patients aged 0–18 years admitted to the consultation clinic or hospitalized to the Children’s Memorial Health Institute over the course of more than 5 years. From this group we separated patients using the available disposable results of 25(OH)D, Ca, PTH (in serum) concentrations.

It was again a wide and diverse population, which included patients admitted for the first time as well as those who were on a subsequent visit to the clinic or undergoing a repeat hospitalization.

The reference point for the statistical evaluation was the available value of 25(OH)D concentration noted during a visit to the consultation clinic. 

The Children’s Memorial Health Institute uniquely takes care of pediatric patients suffering from almost every disorder or disease, therefore, the types of suspected or diagnosed diseases or disorders were at that time diverse. It was assumed that the patients under assessment were supplemented with cholecalciferol according to the current recommendations [35,36]. In addition, it was assumed that the vitamin D supplementation guidelines [35,36], which were recently updated [37], are the mostly followed by parents or caregivers of neonates and infants up to 18 months of age.

The comparison of the mean concentrations of 25(OH)D in the recent study group showed a fairly clear negative correlation in the age groups 0–1 year and 1–3 years. The average values of 25(OH)D concentration were 36.3 ng/mL and 33.6 ng/mL, respectively. Among these age groups, 25(OH)D concentrations remained, in general, within the optimal concentration ranges of 30–50 ng/mL [35,36,37]. There was a significant negative correlation between 25(OH)D concentration and the overall age of patients in the range 0–18 years. However, in the groups of toddlers and younger children aged 3–10 years, as well as in older children and teenagers aged 10–18 years, the average 25(OH)D concentrations did not reach the optimal values of 30–50 ng/mL and were 28.7 ng/mL and 25.4 ng/mL, respectively.

This observation indirectly confirms the widespread use of anti-rickets prophylaxis by pediatricians in the group of the youngest children under the age of 3 years and is in agreement with our previous findings [49]. Further, a strong negative correlation was observed between 25(OH)D concentrations and children of ages 6 and 12 months caused by rapid weight gain which was hypothesized as a reason to increase the supplementation dose of cholecalciferol in infants [53].

The observed changes in the 25(OH)D concentration values within different time periods showed variable average 25(OH)D concentrations from the optimal concentration range of 31.2 ng/mL noted in 2014 to 29 ng/mL noted in 2015–2017, and again in the concentration of 32.1 ng/mL revealed in 2018–2019. 

Recommendations for cholecalciferol supplementation in individual pediatric age groups were not changed within the period of our study, so the above-mentioned slight fluctuations in average 25(OH)D concentrations can be explained by the diversity of disease etiology and or by incomplete compliance with supplementation recommendations.

A comparison of the 25(OH)D concentrations in individual quarters of the year confirms the significant role of skin synthesis in the supply of vitamin D. The visible effect of skin synthesis was revealed especially during the summer months, and it also persisted in autumn months. However, it should be noted that the average concentrations of 25(OH)D in the second and third quarters of the year only slightly exceeded the lower limit of optimal concentrations, approximately 31.3 ng/mL and 30.6 ng/mL, respectively.

Of note, in our laboratory, 25(OH)D concentration measurements were carried out using an automatic IDS-ISYS system based on chemiluminescence immunoassay. 

This method was compared to and validated with the Good Laboratory Practice requirements, and the high precision of 25(OH)D measurements was confirmed with a DEQAS international quality certificate.

The analyses of relations between calcemia and vitamin D status provided interesting observations: more than 72% of patients with 25(OH)D concentration <10 ng/mL (severe vitamin D deficiency) had a normal serum Ca concentration (2.25–2.65 mmol/L). In the subgroup with hypocalcemia (Ca < 2.25 mmol/L), 25% of patients had 25(OH)D concentration <20 ng/mL and 57% had 25(OH)D concentration <30 ng/mL. In the analyzed subgroup with hypercalcemia (Ca > 2.65 mmol/L), 25(OH)D concentrations were both very low and high and there was no correlation between the severity of hypercalcemia and 25(OH)D concentration. Despite hypercalcemia, more than 30% of patients did not achieve optimal 25(OH)D concentration values of at least 30 ng/mL. Moreover, a number of cases with hypercalcemia had 25(OH)D concentrations lower than 10 ng/mL or higher than 50 ng/mL. These results generally confirm a lack of clear correlation between 25(OH)D and Ca serum concentrations and indirectly showed that the causes of disturbances in the Ca-P metabolism may have various causes. 

The evaluation of the active metabolite 1,25(OH)_2_D showed that more than 54% of the estimated population had a concentration above 80 pg/mL; in clinical practice this is considered suitable for further detailed analysis [45,54,55].

High concentrations of calcitriol can also indicate other disorders causing disturbances in Ca-P metabolism, but due to very limited number of cases the further speculations in this case were omitted by us.

The studies conducted in various sites of the world and in Poland have already uncovered vitamin D status in the general population, including pediatric patients, and confirmed the problem of low 25(OH)D concentrations in both healthy and diseased populations [22,23,24,25,27,29,33,56,57]. 

Nonetheless, more research focused on vitamin D and its metabolites is still required, and our study results, due to its limitations, provided only the estimation of negative trends of 25(OH)D concentration with age. The evaluation of the pediatric population covering over 5 years can, in our opinion, provide information not only to the past and current state and trends in the vitamin D status but also to the relation between calcium and 25(OH)D concentrations, at least in 0–18 year olds residing in Poland. To conclude, vitamin D deficiency (<20 ng/mL) was noted in 18.9% subjects in 2014–2019. The effective prevention of vitamin D deficiency was observed in children aged 3 years and younger. A relationship between the concentrations of calcium and 25(OH)D was not observed.

## Figures and Tables

**Figure 1 nutrients-15-03477-f001:**
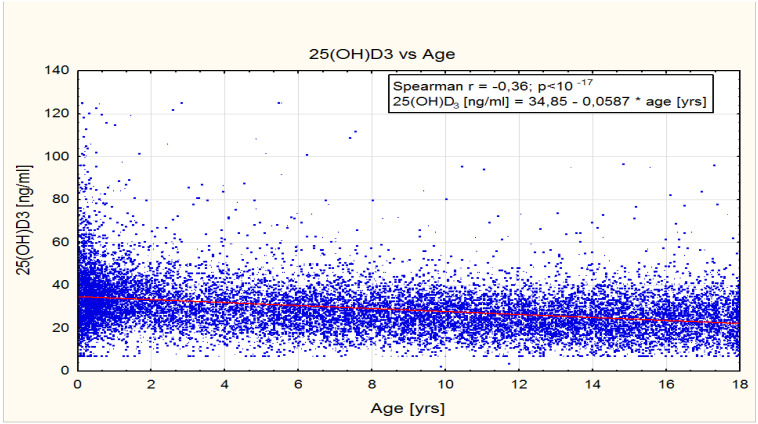
Correlation between 25(OH)D concentrations (ng/mL) and age (years).

**Table 1 nutrients-15-03477-t001:** General characteristics of assessable biochemical parameters in the group of pediatric patients commissioned for calcium –phosphate and vitamin D status evaluations.

Parameter	Number of Participants (*n*)	Mean (SD)	Reference Values
Ca mmol/L	26,731	2.42 (0.16)	2.25–2.65 mmol/L
25(OH)D ng/mL	17,636	29.45 (11.66)	30–50 ng/mL
PTH pg/mL	2839	29.72 (67.32)	11–62 pg/mL
Mean age (months)	49,424	87.17 (71.46)	

**Table 2 nutrients-15-03477-t002:** Distribution of the concentrations of selected biochemical parameters.

**Ca; mmol/L**	**%**	**N**
<2.25	1.04	2797
2.25–2.65	84.6	22,609
2.66–2.75	4.0	1064
≥2.76	1.0	263
**25(OH)D; ng/mL**	**%**	**N**
<10	1.7	292
10–20	17.2	3039
>20–30	39.5	6960
>30–50	37.2	6567
>50–100	4.2	744
>100	0.2	34
**PTH; pg/mL**	**%**	**N**
<11	13.5	382
>11–62	82.3	2330
>62	4.2	118

Ca in serum: min. 0.89 mmol/L, max. 4.82 mmol/L; %—the percent of samples; N—measured samples (patients); 25(OH)D in serum: min. 1.9 ng/mL, max. 125 ng/mL; %—the percent of samples; N—measured samples (patients); PTH in serum: min. 0.72 pg/mL, max. 1515 pg/mL; %—the percent of samples; N—measured samples (patients).

**Table 3 nutrients-15-03477-t003:** Distribution of 25(OH)D concentrations (*n* = 93) among subgroup with evident hypercalcemia (Ca ≥ 2.75).

25(OH)D; ng/mL	%	N	Mean (SD)
<10	2.2	2	8.0 (1.27)
10–20	8.6	8	15.8 (2.20)
>20–30	22.6	21	25.9 (3.04)
>30–50	51.6	48	37.7 (5.36)
>50–100	14.0	13	62.2 (11.00)
>100	1	1	125.0 (0.0)

%—the percent of samples; N—measured samples (patients).

**Table 4 nutrients-15-03477-t004:** Distribution of 1,25(OH)_2_D concentrations (*n* = 37) among subgroup with evident hypercalcemia (Ca ≥ 2.75).

1,25(OH)_2_D; pg/mL	%	N	Mean (SD)
<30	13.5	5	15.1 (9.03)
30–80	32.4	12	58.2 (11.77)
>80	54.1	20	125.6 (36.74)

%—the percent of samples; N—measured samples (patients).

**Table 5 nutrients-15-03477-t005:** Distribution of PTH concentrations (*n* = 41) among subgroup with evident hypercalcemia (Ca ≥ 2.75).

PTH; pg/mL	%	N	Mean (SD)
<11	53.7	22	4.2 (3.09)
11–62	41.5	17	22.7 (10.21)
>62	0.8	2	475.4 (491.43)

%—the percent of samples; N—measured samples (patients).

**Table 6 nutrients-15-03477-t006:** Mean 25(OH)D concentrations in subsequent age groups.

Age (Months)	Age (Years)	N	25(OH)D ng/mL (SD)
0–12	0–1	3483	36.3 (14.15)
>12–36	>1–3	1917	33.6 (19.76)
>36–120	>3–10	5382	28.7 (10.25)
>120–216	>10–18	6908	25.4 (9.11)

N—measured samples (patients).

**Table 7 nutrients-15-03477-t007:** Mean 25(OH)D concentration values calculated in respective time periods during 6 years of operation of single diagnostic unit.

Time Period (Years)	N	Mean Age (Months, Years)	25(OH)D ng/mL (SD)
2014	6045	77.7 (6.5)	31.2 (12.54)
2015	6738	82.2 (6.9)	27.2 (10.97)
2016	15,551	94.1 (7.8)	28.7 (10.46)
2017	9771	88.8 (7.4)	28.3 (11.31)
2018–19	11,319	84.2 (7.0)	32.1 (12.85)

N—measured samples (patients).

**Table 8 nutrients-15-03477-t008:** 25(OH)D concentrations in relation to quarter of the year.

Quarter of Year	N	25(OH)D ng/mL (SD)
I	4885	27.9 (12.18)
II	3815	31.3 (10.91)
III	4540	30.6 (10.97)
IV	4396	28.3 (12.30)

N—measured samples (patients).

## Data Availability

Not applicable.

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
