# Peer review of "25(OH)D Concentration in Neonates, Infants, Toddlers, Older Children and Teenagers from Poland—Evaluation of Trends during Years 2014–2019"

_nutrients, 2023, doi:10.3390/nu15153477_

Round 1
Reviewer 1 Report
Major comment:
The authors present data on serum levels of 25(OH)D, calcium and parathyroid hormone in a pediatric patient population from a single center in Poland.
The value of the article lies in the large number of patients that allows statistical conclusions. The authors did try statistical analysis of correlation between 25(OH)D and age. It would be interesting to calculate possible regression between 25(OH)D and PTH and calcium.
A discrepancy is that calcium levels are given in mmol/L while 25(OH)D levels are given in ng/mL and PTH in pg/mL. The authors should either use mol units for all variables or present calcium values using the most common unit mg/dL.
Minor comments:
Lines 119-122: The description of the methods needs improvement. Instead of a generic description of photometry, or IRMA the authors should refer to the principles and details of the specific methods that they used.
Lines 142-156. The statistical package used is not mentioned.
Average quality
Author Response
Reviewer #1
Major comment:
The authors present data on serum levels of 25(OH)D, calcium and parathyroid hormone in a pediatric patient population from a single center in Poland.
The value of the article lies in the large number of patients that allows statistical conclusions. The authors did try statistical analysis of correlation between 25(OH)D and age. It would be interesting to calculate possible regression between 25(OH)D and PTH and calcium.
A discrepancy is that calcium levels are given in mmol/L while 25(OH)D levels are given in ng/mL and PTH in pg/mL. The authors should either use mol units for all variables or present calcium values using the most common unit mg/dL.
Author’s response: We would like to thank Reviewer 1 for valuable suggestions and overall comment. The correlation between Ca and 25(OH)D were not statistically significant, however this statement was added to Results section, despite that was previously in Discussion section – thank you for your suggestion. Unfortunately, due to number of samples of 25(OH)D we were not able to convert the results to nmol/L, sorry for that.
Minor comments:
Lines 119-122: The description of the methods needs improvement. Instead of a generic description of photometry, or IRMA the authors should refer to the principles and details of the specific methods that they used.
Lines 142-156. The statistical package used is not mentioned.
Author’s response: These suggestions were implemented in the text. The methods section was wrote again with statistical package added (Statistica v11 or 10).
Reviewer 2 Report
In the manuscript submitted to me for review entitled: “25(OH)D Concentration in Neonates, Infants, Toddlers, Older Children and Teenagers From Poland—Evaluation of Trends During Years 2014-2019” the authors studied the trends in the change of the serum concentration of 25-hydroxyvitamin D (25(OH)D) in the period 2014 - 2019 in children aged 0 - 18 years.
In the introduction, the authors present the accepted optimal amounts for vitamin D intake, as well as the negatives of its deficiency or overdose.
In the „Methods“ section, my opinion is that the methods are not well presented. Only the meaning of the methodology and what information it brings us is described, but the actual implementation of the method in steps and the materials used are not described. Everything is described very generally, and if another team wants to repeat the experiments according to this description, they will not be able to.
The results are well described and presented using 8 tables and 1 figure. In support of their research, the authors used 59 references, of which 12 are from the last 5 years (1/5 of the total number).
The research carried out is very extensive and consistent and I express my admiration for the work of the authors. This is a very valuable study which I suspect will interest the reader.
I have some remarks and recommendations to the authors that I think will improve the quality of the manuscript.
1. To describe the methods in a little more detail.
2. In Table 1, why is the last column "participants" completely empty?
3. Table 2 does not clearly indicate the meaning of the presented values. For example "%" - this symbol by itself means nothing. Let the % of what be indicated. It is also not indicated what "N" means - the meaning of the symbol can be indicated with a * sign below the table. The same remark applies to Tables 3, 4, 5, 6, 7 and 8.
4. In the caption of Table 5, there is a spelling error in the word "hypercalcaemia". Let it be corrected.
5. In general, the "Conclusion" section is optional. The authors have made a nice discussion of about 2 pages, but in my opinion the results are many and after the discussion they will be better summarized if they are described in 2-3 sentences in Conclusion.
6. At the end of the manuscript before the References section, authors have not submitted the mandatory Author Contributions section. In the Conflict of interest section, they must state that they declare that they have no conflict of interest.
7. In the References section, the literary sources with the following numbers are presented not with all authors: 4, 5, 7, 15, 16, 17, 20, 21, 23, 24, 28, 29, 30, 35, 36, 37, 38 , 39, 40, 41, 42, 43, 51, 52, 53 and 59. Let all authors be added.
Author Response
Reviewer #2
In the manuscript submitted to me for review entitled: “25(OH)D Concentration in Neonates, Infants, Toddlers, Older Children and Teenagers From Poland—Evaluation of Trends During Years 2014-2019” the authors studied the trends in the change of the serum concentration of 25-hydroxyvitamin D (25(OH)D) in the period 2014 - 2019 in children aged 0 - 18 years.
In the introduction, the authors present the accepted optimal amounts for vitamin D intake, as well as the negatives of its deficiency or overdose.
In the „Methods“ section, my opinion is that the methods are not well presented. Only the meaning of the methodology and what information it brings us is described, but the actual implementation of the method in steps and the materials used are not described. Everything is described very generally, and if another team wants to repeat the experiments according to this description, they will not be able to.
The results are well described and presented using 8 tables and 1 figure. In support of their research, the authors used 59 references, of which 12 are from the last 5 years (1/5 of the total number).
The research carried out is very extensive and consistent and I express my admiration for the work of the authors. This is a very valuable study which I suspect will interest the reader.
Author’s response: We would like to thank Reviewer 2 for valuable suggestions and overall comments. Indeed the methods section and references were not so well provided. After your suggestion we have write the method section again, adding the statistics. References, although rather old, are the same due to the fact that similar, most recent extensive studies for Poland, are not known by us.
I have some remarks and recommendations to the authors that I think will improve the quality of the manuscript.
- To describe the methods in a little more detail.
- In Table 1, why is the last column "participants" completely empty?
- Table 2 does not clearly indicate the meaning of the presented values. For example "%" - this symbol by itself means nothing. Let the % of what be indicated. It is also not indicated what "N" means - the meaning of the symbol can be indicated with a * sign below the table. The same remark applies to Tables 3, 4, 5, 6, 7 and 8.
- In the caption of Table 5, there is a spelling error in the word "hypercalcaemia". Let it be corrected.
- In general, the "Conclusion" section is optional. The authors have made a nice discussion of about 2 pages, but in my opinion the results are many and after the discussion they will be better summarized if they are described in 2-3 sentences in Conclusion.
- At the end of the manuscript before the References section, authors have not submitted the mandatory Author Contributions section. In the Conflict of interest section, they must state that they declare that they have no conflict of interest.
- In the References section, the literary sources with the following numbers are presented not with all authors: 4, 5, 7, 15, 16, 17, 20, 21, 23, 24, 28, 29, 30, 35, 36, 37, 38 , 39, 40, 41, 42, 43, 51, 52, 53 and 59. Let all authors be added.
Author’s response: We would like to thank Reviewer 2 for the comments, your suggestions were implemented in the text. Author Contributions were added, the errors of hypercalcaemia were done (hypercalcemia), the note to “N” and”%” were added to tables as well as “participants” were corrected – thank you for your time spent to read our paper and to provide us your feedback.